# The Association of Cooking Fuel Use, Dietary Intake, and Blood Pressure among Rural Women in China

**DOI:** 10.3390/ijerph17155516

**Published:** 2020-07-30

**Authors:** Alexandra L. Bellows, Donna Spiegelman, Shufa Du, Lindsay M. Jaacks

**Affiliations:** 1Department of Global Health and Population, Harvard T.H. Chan School of Public Health, Boston, MA 02115, USA; alexandra.bellows@jhu.edu; 2Center for Methods on Implementation and Prevention Science, Yale School of Public Health, New Haven, CT 06520, USA; donna.spiegelman@yale.edu; 3Department of Nutrition, The University of North Carolina at Chapel Hill, Chapel Hill, NC 27599, USA; dushufa@email.unc.edu; 4Global Academy of Agriculture and Food Security, The University of Edinburgh, Edinburgh EH25 9RG, UK

**Keywords:** indoor air pollution, nutrition, Asia, cohort study

## Abstract

Household air pollution (HAP) from solid cooking fuels continues to affect 600 million people in China and has been associated with high blood pressure. The role of diet in HAP-associated high blood pressure has yet to be evaluated in China. The aim of this study was to estimate the impact of cooking fuel on change in blood pressure and evaluate whether intake of antioxidant- and omega-3 fatty acid-rich foods (fruits, vegetables, and seafood) attenuates any adverse effects of solid fuel use on blood pressure. We analyzed longitudinal data collected between 1991 and 2011 from nonpregnant women aged 18 to 80 years living in rural areas of China. We used linear mixed effects models to estimate the association between cooking fuel (coal or wood versus clean [electric or liquid petroleum gas]) and blood pressure. Possible mediation of the fuel effect by diet was assessed by the difference method. A total of 6671 women were included in this study. Women less than 40 years of age cooking with cleaner fuels over time had lower rates of change in systolic blood pressure compared to women cooking with coal (*p* = 0.004), and this effect was not mediated by dietary intake. Associations between fuel use and change in diastolic blood pressure were not significant. These findings lend further support for there being a direct effect of reducing HAP on improvements in blood pressure, independent of concurrent dietary intake.

## 1. Introduction

In 2016, an estimated 2.6 million premature deaths worldwide were attributable to household air pollution (HAP) [1,2,3]. Approximately one-quarter of these deaths (23%) occurred in China, where, according to results of a national survey conducted in 2012, 13.5% and 21.5% of rural households use coal and 47.6% and 19.0% use biomass (mainly wood and crop residues) for cooking and heating, respectively [4]. The use of solid fuel in rural China has been found to be associated with increased risk of cardiovascular disease (CVD) and all-cause mortality [5]. The underlying biological mechanism linking HAP exposure to CVD may involve the promotion of oxidative stress pathways by fine particulate matter less than 2.5 µm in diameter (PM_2.5_), which in turn leads to vasoconstriction, thereby increasing blood pressure—an important CVD risk factor [6]. Both cross-sectional and longitudinal observational studies have reported a significant positive association between HAP and blood pressure in China [7,8,9,10,11].

At the same time, China is undergoing a significant nutrition transition away from diets high in vegetables, legumes, and whole grains to diets high in animal-based products, refined grains, and salt [12]. Numerous studies have found significant associations between these emerging unhealthy dietary patterns and high blood pressure in China [13,14]. However, to the best of our knowledge, no study has evaluated the role of dietary intake in HAP-associated high blood pressure in China. Recently, a cross-sectional analysis of 205 rural women in Sichuan, China found a borderline significant positive association between PM_2.5_ exposure and blood pressure among those 50 years or older after adjusting for sodium intake and other CVD risk factors (e.g., second-hand tobacco smoke exposure and alcohol intake) but did not explore other dietary factors [11]. In that study, the effects among younger women were nonsignificant [11], which has been observed in previous studies in China and Guatemala [9,15].

Previous studies focused on ambient air pollution in high-income countries have shown that high dietary intake of antioxidants may attenuate PM_2.5_-related high blood pressure and CVD [16,17,18]. Moreover, a small randomized, controlled study demonstrated a protective effect of fish oil supplements on heart rate variability and electrocardiographic repolarization changes resulting from exposure to concentrated ambient fine and ultrafine particulate matter [19].

Using data from rural women participating in the prospective China Health and Nutrition Survey (CHNS) cohort, the aims of this study were: (1) to estimate the prospective association of cooking fuel use with blood pressure and (2) to determine the proportion of this association mediated by dietary intake. We hypothesized that (1) the use of coal or wood cooking fuel would be associated with greater increases in systolic and diastolic blood pressure compared to clean cooking fuel (gas or electric), especially among older adult women and (2) higher intakes of antioxidant- and omega-3 fatty acid-rich foods (fruits, vegetables, and seafood) would attenuate any adverse effects of solid cooking fuel on blood pressure. This would be the first study in China to assess if diet may mediate the relationship between HAP and high blood pressure.

## 2. Materials and Methods

### 2.1. Study Population

CHNS is an ongoing longitudinal study of a demographically and socioeconomically diverse cohort from nine provinces in China: Guangxi, Guizhou, Heilongjiang, Henan, Hubei, Hunan, Jiangsu, Liaoning, and Shandong [20]. Data from CHNS are available online at: http://www.cpc.unc.edu/projects/china. Questionnaires and anthropometric data were collected in 1989, 1991, 1993, 1997, 2000, 2004, 2006, 2009, and 2011. Due to differences in eligibility criteria and consistent with many previous analyses of CHNS, the 1989 survey was excluded. This study was limited to nonpregnant women aged 18 to <80 years living in rural areas who reported their primary source of cooking fuel as either electric, liquid petroleum gas (LPG), coal, or wood and had systolic blood pressure (SBP) measured. We focused on women because women typically have a higher exposure to HAP owing to the fact that they are responsible for the majority of cooking in this context. Additionally, previous studies have found a high prevalence of smoking among men in this population (>50% in rural areas), which would confound the relationship between HAP and blood pressure [21]. Women taking hypertension drugs were excluded. In addition, women were excluded if they were missing baseline (e.g., first observation in that participant’s series) income, urbanicity index, level of education, alcohol consumption, or smoking status. “Rural” areas were defined as areas in China with a population less than 100,000 [22]. A summary of the number of CHNS participants that met each inclusion and exclusion criterion is provided in Appendix A.

Data collection procedures were approved by Institutional Review Boards of the University of North Carolina, Chapel Hill (Study #: 07-1963) and the Chinese Center for Disease Control and Prevention, and all participants gave written informed consent.

### 2.2. Data Collection

With regards to the cooking fuel assessment, the household survey included a question, “What kind of fuel does your household normally use for cooking?” Prespecified options included coal, electricity, kerosene, LPG, natural gas, wood, charcoal, and other. In this study, “clean fuel” was defined as those using LPG or electricity, which is consistent with the 2014 WHO Guidelines for indoor air quality. If households used multiple fuel sources for cooking, participants were asked to list the fuel they used most often as the primary fuel and identify other fuel they used as the secondary fuel source.

Individual dietary intake was assessed at each survey round over a three-day period using three consecutive 24-h recall surveys. Individuals were asked to report all foods they consumed outside of the home and all foods consumed at home. To help estimate portion size, trained enumerators presented food models and picture aids to participants during 24-h recall assessments. For mixed dishes consumed at home, enumerators asked for additional information regarding the amount of each ingredient used in the recipe. Individuals within each household were asked to report the estimated proportion of each mixed dish they consumed. Foods consumed at home were checked against household food consumption data that were collected during the same three days as the individual 24-h recall [23]. From 1991 to 2009, this involved weighing all ingredients at the household level and the amount remaining after consumption. Disappearance values were compared to values reported by the individual. When inconsistencies were identified, field staff revisited each individual’s food consumption data. In 2011, this procedure was only done for condiments (oil, salt, monosodium glutamate, etc.). Food groups were converted to grams/1000 kcal/day. Dietary intake was restricted to those who reported consuming a three-day average of greater than 500 kcals and less than 5000 kcals.

Physicians with additional training measured blood pressure of participants using mercury sphygmomanometers. After resting for at least 5 min, systolic and diastolic blood pressure (SBP and DBP) were measured in triplicate with at least one minute in between measurements while the participant was in a seated position. Mean SBP and DBP were calculated as the average of the second and third measurements [24].

Potential confounders evaluated included age, baseline annual total household income, community urbanicity index, education level, smoking status, alcohol intake, and baseline body mass index (BMI). Annual total household income was a composite variable in the dataset constructed from nine sources of potential income: business, farming, fishing, gardening, livestock, nonretirement wages, retirement income, subsidies, and other. Community urbanicity index was a composite variable in the dataset constructed from 12 indicators including: population density, economic activity, traditional markets, modern markets, transportation infrastructure, sanitation, communications, housing, education, diversity, health infrastructure, and social services [25]. Education level was categorized as less than primary school, completed primary school, some high school, and high school degree or higher. Smoking status was defined as having ever smoked cigarettes (yes/no). Alcohol intake was defined as either never consumed alcohol or consumed alcohol on frequent or infrequent occasions. Baseline BMI was calculated from height and weight measured by trained enumerators using standardized protocols.

### 2.3. Statistical Analysis

We first present descriptive statistics of demographic and socio-economic characteristics by cooking fuel category. Three categories were defined as follows: women who reported using electricity or LPG as their primary source of cooking fuel, women who reported using coal as their primary source of cooking fuel, and women who reported using wood as their primary source of cooking fuel. We decided to differentiate between wood and coal users given that the composition of emissions differs by fuel type [26,27].

We then used longitudinal linear mixed effects models under an exchangeable working correlation [28] to estimate the association between cooking fuel use and blood pressure, with clustering set at the highest level of nesting, the household [29,30]. We used stepwise restricted cubic spline models [31,32] to evaluate any possible nonlinear associations between age and blood pressure and found an inflection point around age 40. We thus included the interaction between age and all other covariates in the models.

Because the trajectory of blood pressure changes over age appeared to be nonlinear with an inflection point at age 40, the final models were stratified by age: less than 40 or greater than or equal to 40 years. Final models included fuel category, age (years), fuel category*age, survey year (categorical, reference = 2011), survey year*age, baseline income quintiles (categorical, reference = 3rd quintile), baseline income quintiles*age, urbanicity index quintiles (categorical, reference = 3rd quintile), urbanicity index*age, education level (categorical, reference = “some primary level education”), education level*age, alcohol intake (consumer/not consumer, reference = not consumer), alcohol intake*age, ever smoked (yes/no, reference = no), ever smoked*age, baseline BMI (kg/m^2^), and baseline BMI*age. Models that controlled for dietary intake included consumption of fruits, vegetables, and seafood, all in g/1000 kcal/day. We checked for multicollinearity among dietary components by calculating the Pearson correlation coefficient among the three food groups. These food groups were selected based on findings from previous studies suggesting that they may attenuate the adverse effect of air pollution on blood pressure [16,17,18,19]. Missingness for BMI and dietary intake was addressed by imputing the median of these variables at each survey year and including a missing indicator variable in the model.

In order to visualize the magnitude of estimated effects, we plotted predicted values for SBP and DBP by age for each source of cooking fuel. To obtain predicted values, we used the predict command in Stata v15 (StataCorp LLC, College Station, TX, USA) to apply predicted values obtained from longitudinal models to datasets where age varied from 18–<40 and 40–80 years. Other covariates in the model besides fuel category were set to reference values for categorical variables and median values for continuous variables. The proportion mediated by diet was estimated by calculating the percent difference in the cooking fuel effects for models with and without dietary intake [33]. Analyses were conducted using SAS v9.4 (SAS Institute, Cary, NC, USA) and Stata v15 (StataCorp LLC, College Station, TX, USA).

### 2.4. Sensitivity Analysis

We conducted a sensitivity analysis using the same models described above to see if our results were confounded by solid fuel use as a secondary source of fuel. In this analysis, we excluded households who reported primary fuel use as either electric or LPG but secondary fuel use as coal or wood.

## 3. Results

A total of n = 6671 women from 4169 households met the inclusion criteria for this analysis. Overall, there were 22,118 observations. The number of observations per woman ranged from 1 to 8 (median of 3), with 5.6% (n = 375) of women having 8 observations and 31.4% (n = 2096) of women having only 1 observation.

From 1991 to 2006, the majority of women reported using either wood or coal as cooking fuel at home. After 2006, over 50% of households reported using clean fuel for cooking: 41% of women in 2006, 58% of women in 2009, and 65% of women in 2011 (Figure 1). On average, women who used clean fuel came from households with higher household incomes and higher urbanicity index scores and attained higher levels of education compared to women using wood or coal as cooking fuel (Table 1). Women who used wood or coal spent more time cooking and had lower BMIs compared to women using clean fuel. Women who used clean fuel consumed more meat, seafood, fruit, and oil, and less salt and rice than women cooking with wood or coal. The correlation coefficient between seafood and vegetable consumption was 0.02. The correlation coefficient between seafood and fruit consumption was 0.08, and for vegetable and fruit consumption was −0.02.

For women <40 years old, SBP increased at a faster rate for those using coal as their primary cooking fuel compared to women using clean fuel (*p* = 0.004), but for women ≥40 years of age we saw a similar rate of increase in SBP across the three groups (Figure 2). The average difference in slope for predicted SBP between coal and clean cooking fuel in women <40 years was 0.16 mm Hg per year of age (95% CI: 0.05, 0.27), and for women ≥40 years, it was 0.07 mm Hg per year of age (95% CI: −0.02, 0.15). The average difference in slope for predicted SBP between wood and clean cooking fuel in women <40 years was 0.04 mm Hg per year of age (95% CI: −0.09, 0.16), and for women ≥40 years, it was 0.02 mm Hg per year of age (95% CI: −0.06, 0.11). The proportion of the association between coal and change in SBP attenuated by dietary intake was less than 1% across all ages (Table 2).

For women <40 years old and ≥40 years old, DBP increased at a slightly faster rate for women using coal (<40 years: *p* = 0.07, ≥40 years: *p* = 0.09) (Figure 3). The average difference in slope for predicted DBP between coal and clean cooking fuel in women <40 years was 0.08 mm Hg per year of age (95% CI: −0.005, 0.17), and for women ≥40 years, it was 0.05 mm Hg per year of age (95% CI: −0.008, 0.10). The average difference in slope for predicted DBP between wood and clean cooking fuel in women <40 years was 0.01 mm Hg per year of age (95% CI: −0.08, 0.11) and for women ≥40 years it was 0.009 mm Hg per year of age (95% CI: −0.04, 0.06). The proportion of the association between coal and change in DBP attenuated by dietary intake was less than 1% across all ages (Table 2).

We found that 2787 of the 7114 (39%) participants who reported clean fuel as their main cooking fuel source also reported secondary use of either coal or wood for cooking fuel. When we excluded these participants from our analysis, we found the average difference in slope for predicted SBP between coal and clean cooking fuel in women <40 years was 0.20 mm Hg per year of age (95% CI: 0.07, 0.33), and for women ≥40 years, it was 0.08 mm Hg per year of age (95% CI: −0.02, 0.19). The average difference in slope for predicted SBP between wood and clean cooking fuel in women <40 years was 0.08 mm Hg per year of age (95% CI: −0.07, 0.22), and for women ≥40 years, it was 0.03 mm Hg per year of age (95% CI: −0.08, 0.14). The average difference in slope for predicted DBP between coal and clean cooking fuel in women <40 years was 0.08 mm Hg per year of age (95% CI: −0.03, 0.18) and for women ≥40 years it was 0.03 mm Hg per year of age (95% CI: −0.04, 0.09). The average difference in slope for predicted DBP between wood and clean cooking fuel in women <40 years was 0.01 mm Hg per year of age (95% CI: −0.10, 0.12) and for women ≥40 years it was −0.01 mm Hg per year of age (95% CI: −0.08, 0.06).

## 4. Discussion

Data from over 20 years of observation in this prospective cohort show that women in rural China increasingly cooked with cleaner fuels (LPG or electric), especially those with higher socio-economic status. Women cooking with cleaner fuels over time had marginally smaller increases in SBP and DBP compared to women using coal, and this effect was not attenuated by dietary intake. Together, these findings lend further support to a direct effect of HAP on blood pressure.

We found that in unadjusted analyses, women who used wood or coal had lower SBP and DBP than women using clean fuel, likely due to higher levels of household income, urbanicity, and higher BMI among clean fuel users. These findings are consistent with a recent analysis of 12 Demographic and Health Surveys (DHS) from 10 countries [34]. The effects we observed—about a 0.16 mm Hg difference in SBP (*p* < 0.01) and 0.08 mm Hg difference in DBP (*p* < 0.10) among women <40 years using coal compared to clean fuel and 0.07 mm Hg difference in SBP (*p* > 0.10) and 0.05 mm Hg difference in DBP (*p* < 0.10) among women ≥40 years using coal compared to clean fuel—were lower in terms of magnitude with previous studies quantifying the effect of HAP on blood pressure. For example, in the aforementioned analysis of DHS data (women aged 15–49 years), the adjusted difference in SBP between solid fuel users and clean fuel users was 0.58 mm Hg [34]. Similarly, studies among women in China (SBP change <0.8 mm Hg for 1-unit increase in the log of PM_2.5_) [9] and Ghana (within-subject change in SBP of −2.1 mm Hg among women receiving an improved cookstove compared to controls) [35] have found modest effects on blood pressure.

Only a few studies have assessed the effects of HAP on blood pressure stratified by age group. In contrast to our finding, those studies found a significant effect in older women but no effect in younger women [9,36]. Both of those studies directly measured PM_2.5_ exposure instead of using a proxy measure. Additionally, another reason we may see differences in magnitude of effect compared to prior studies is the fact that we were assessing change in blood pressure over time as opposed to differences in blood pressure at a single time point.

There are several reasons that may explain why larger differences in blood pressure were not observed. For example, while we found that women cooking with wood or coal spent significantly longer cooking (about 2 h more per day) compared to women cooking with clean fuel, a recent study in Sichuan, China found that longer traditional (biomass) stove use was not associated with higher indoor PM_2.5_ levels [37]. In that same study, they found that only 24% of days with exclusive use of clean fuel stoves met the WHO indoor air quality target of 35 µg/m^3^ for PM_2.5_, suggesting an important role of local outdoor air pollution [37]. However, a recent analysis found that 90% of reductions in population-weighted exposure to PM_2.5_ in China from 2005 to 2015 could be attributed to reduced household solid-fuel use, especially fuel use for cooking—supporting the critical role of clean fuels in reducing exposure [38]. Another plausible explanation could relate to “stove stacking,” a common practice in which households utilize a combination of clean and solid fuel. The study in Sichuan, China found that 24% of households surveyed reported “mixed” stove use and found that homes with mixed stove usage had the highest levels of PM_2.5_ [37]. In our study, 39% of households who reported clean fuel as their primary fuel source also reported the use of wood or coal as their secondary source of fuel. A sensitivity analysis, excluding these potential stove-stackers, found similar results as the full sample, with only marginally stronger associations with change in SBP among coal and wood users as compared to clean fuel users. Further research is needed in rural areas to understand the relative contribution of indoor versus outdoor air pollution to improve targeting of interventions to reduce PM_2.5_ exposures.

We found that only a small proportion (<1%) of the effect of HAP on blood pressure was mediated by intake of rich sources of antioxidants and omega-3 fatty acids (vegetables, fruits, and seafood). This is consistent with a previous community-based study in Detroit that found large adverse effects of PM_2.5_ on SBP remained after accounting for a small beneficial effect of dietary antioxidant intake [16]. On the other hand, a retrospective study in Hong Kong found that the effects of ambient air pollution (specifically, levels of PM_10_, NO_2_, and O_3_) on all-cause mortality were lower in those who consumed fruit regularly compared to those who seldom or never consumed fruit (excess relative risk per 10 µg/m^3^ increase in air pollutant due to interaction of about −0.50%), but no effect of seafood or vegetable intake [39]. It should be noted that dietary intake of antioxidant- and omega-3-rich foods (vegetables, fruits, and seafood) was below the recommended intake in our sample. The Food Guide Pagoda for Chinese Residents recommends 200–350 g/day for fruits, 300–500 g/day for vegetables, and 40–75 g/day for seafood [40]. Consumption of these nutritious food groups was low among solid fuel users in this sample: average consumption of fruit, vegetables, and seafood was less than 10 g/1000 kcal/day, 150 g/1000 kcal/day, and 10 g/1000 kcal/day, respectively. Therefore, it is possible that levels of antioxidants and omega-3 fatty acids were not sufficient to protect against the adverse impacts of HAP on blood pressure.

A major strength of this analysis was the differentiation of coal and wood, considering most previous studies have only evaluated the combined effects of solid fuels. We found that coal tended to have a more adverse effect on blood pressure than wood, which is not surprising given that coal particulate matter may contain containments such as sulfur, arsenic, lead, and mercury not typically found in particulate matter from wood combustion [26,27]. This is one of the largest long-term longitudinal studies to assess the effect of cooking fuel on blood pressure—including 283,352 person-years. To our knowledge, this is the first study that attempted to analyze how dietary intake affects the relationship between cooking fuel and blood pressure in a developing country setting. Another strength of this study is the thorough measurement of dietary intake among participants. This study is not without limitations, perhaps the most significant of which is that our exposure was a categorical indicator of cooking fuel use, which is an indirect, crude measure of HAP. We did not have information on the type of stove (e.g., traditional versus “improved”), but field studies have indicated that current “improved” stoves fueled with biomass are unlikely to achieve the exposure reductions necessary to achieve meaningful health benefits, and so the focus has shifted to clean fuels rather than improved stoves [41,42,43,44]. We could not account for second-hand tobacco smoke exposure or outdoor air pollution, which would have attenuated our effect estimates. Moreover, while diet was assessed through rigorous methods to reduce bias in the measurement of the amount of food consumed on a consumption day, three days of 24-h recalls may not represent an individual’s usual intake, particularly for foods that are consumed episodically (for example, once a week or only a couple times a month). Model-based approaches such as the National Cancer Institute (NCI) method have been developed to help estimate usual intake of episodically-consumed foods using only a few days of 24-h recalls [45]. For episodically consumed foods, the NCI method involves a 2-part model in which the probability of consumption is first estimated and then the amount. The amount part of the model only includes participants with intakes >0. When a limited number of participants have intakes >0, the adjusted usual intake distribution is likely to be imprecise [46]. For foods with a large number of zero intakes, such as those included in this analysis (seafood, fruit, and vegetables), including mean estimates from a food frequency questionnaire (FFQ) as covariates substantially improves the power to detect relationships [47]. Unfortunately, however, CHNS did not administer an FFQ. Taking all of these factors into consideration, as well as the fact that diet accounted for a negligible amount of the effect of cooking fuel on SBP in mediation analyses (<1%), implementing the NCI method in this analysis would be unlikely to influence the findings or conclusions. Nonetheless, future studies at the intersection of dietary intake and environmental health should consider the implications of this source of measurement error for their research questions and in their particular sample populations. Information on nutritional supplements, which could be an important source of omega-3 fatty acids, was not reported by participants, but we believe that prevalence of supplementation is low in this population given results from a nationally representative survey in China that found supplement use in the rural population to be less than 1% [48]. Finally, while CHNS is a large study covering nine provinces in China, it is not sampled to be nationally representative, and therefore, these findings may not be generalizable to the country as a whole.

The results of this analysis highlight the need to address HAP in China as part of a comprehensive national plan to reduce blood pressure and CVD risk. In Shanghai alone, by 2030, direct health expenditures for CVD are estimated to reach $1.12 billion, which is nearly double the expenditures in 2012 [49]. Importantly, we report for the first time that the effects of HAP on blood pressure are largely independent of concurrent dietary intake. Future studies, particularly randomized controlled trials of clean cookstoves, will be important opportunities to explore this finding further. In the absence of intervention, HAP-attributable CVD threatens to substantially hinder further development in China.

## 5. Conclusions

From 1991 to 2011, women in rural China increasingly cooked with cleaner fuels (LPG or electric), especially those with higher socioeconomic status. Women cooking with cleaner fuels over time had marginally smaller increases in blood pressure compared to women using coal, and only a small proportion of this effect was attenuated by dietary intake. Together, these findings lend further support to a direct effect of HAP on blood pressure and suggest that healthy diets characterized by higher intakes of fruits, vegetables, and seafood may not fully protect against deleterious cardiovascular effects of HAP. Future studies and research are needed on effective interventions and policies that mitigate HAP exposure to prevent CVD in China.

## Figures and Tables

**Figure 1 ijerph-17-05516-f001:**
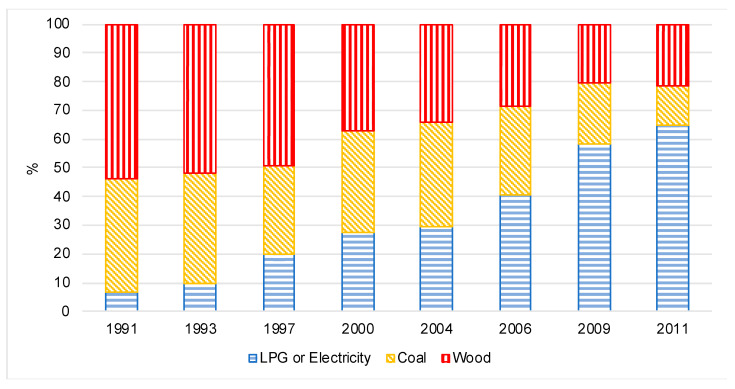
Cooking fuel use across survey years among nonpregnant women aged 18 to <80 years living in rural areas of China (n = 22,118 observations from n = 6671 women, median observations per woman 3).

**Figure 2 ijerph-17-05516-f002:**
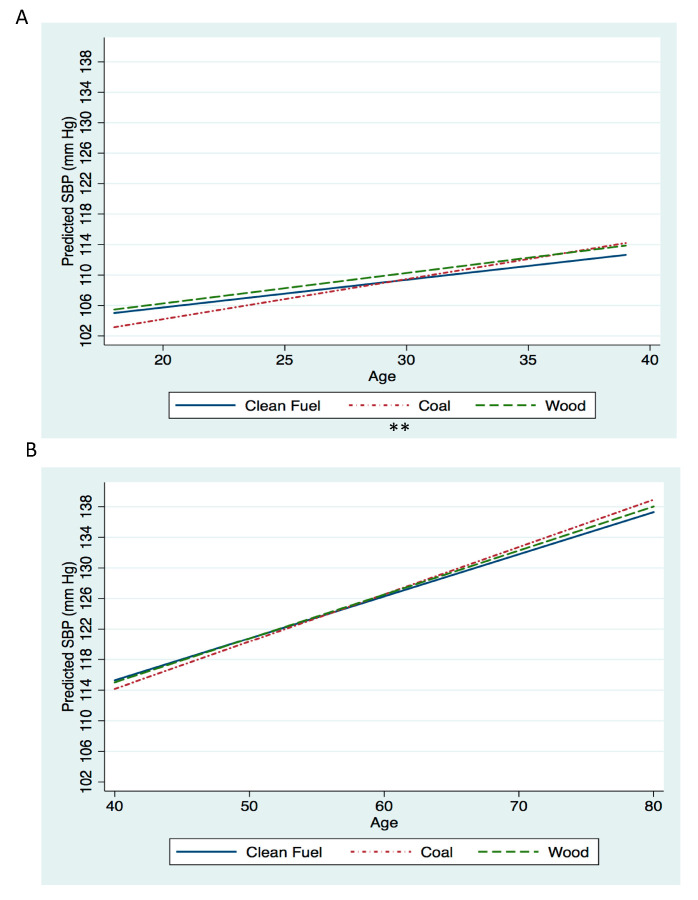
Systolic blood pressure (SBP) by age according to cooking fuel category among nonpregnant women living in rural areas of China (n = 22,118 observations from n = 6671 women, median observations per woman 3). Panel A is the model for women <40 years of age. Panel B is the model for women ≥40 years of age. All models controlled for the following covariates: fuel category, age (years), fuel category*age, survey year (categorical, reference = 2011), survey year*age, baseline income quintiles (categorical, reference = 3rd quintile), baseline income quintiles*age, urbanicity index quintiles (categorical, reference = 3rd quintile), urbanicity index*age, education level (categorical, reference = “some primary level education”), education level*age, alcohol intake (consumer/not consumer, reference = not consumer), alcohol intake*age, ever smoked (yes/no, reference = no), ever smoked*age, baseline BMI (kg/m^2^), and baseline BMI*age. In order to obtain predictive values, variables were set to reference values for categorical variables and median values for continuous variables. ** *p*-value for difference in change in SBP compared to clean fuel <0.01.

**Figure 3 ijerph-17-05516-f003:**
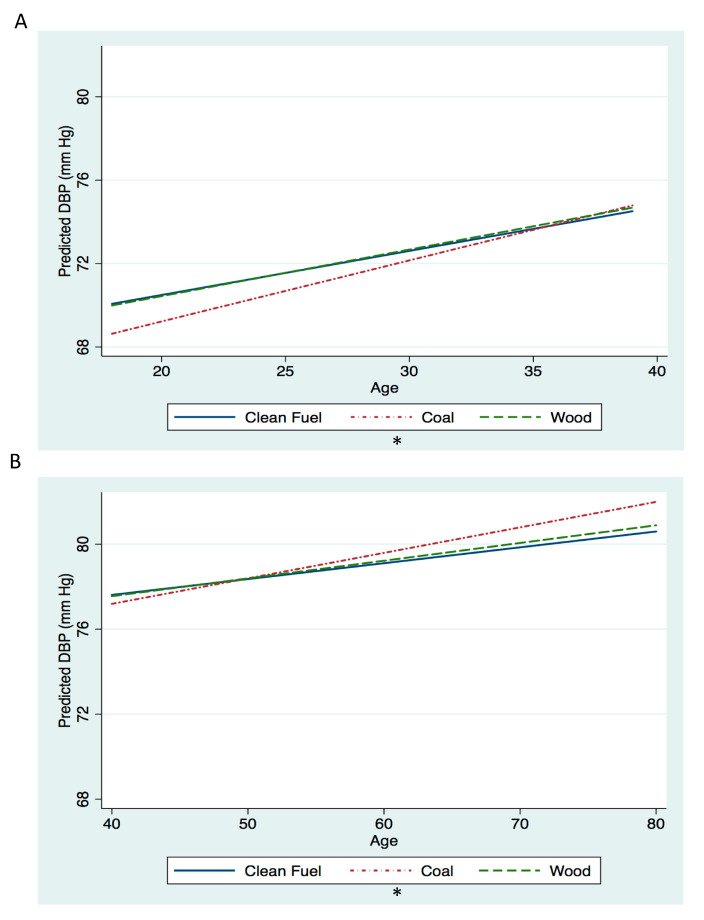
Diastolic blood pressure (DBP) by age according to cooking fuel category among nonpregnant women aged living in rural areas of China (n = 22,118 observations from n = 6671 women, median observations per woman 3). Panel A is the model for women <40 years of age. Panel B is the model for women ≥40 years of age. All models controlled for the following covariates: fuel category, age (years), fuel category*age, survey year (categorical, reference = 2011), survey year*age, baseline income quintiles (categorical, reference = 3rd quintile), baseline income quintiles*age, urbanicity index quintiles (categorical, reference = 3rd quintile), urbanicity index*age, education level (categorical, reference = “some primary level education”), education level*age, alcohol intake (consumer/not consumer, reference = not consumer), alcohol intake*age, ever smoked (yes/no, reference = no), ever smoked*age, baseline BMI (kg/m^2^), and baseline BMI*age. In order to obtain predictive values, variables were set to reference values for categorical variables and median values for continuous variables. * *p*-value for difference in change in DBP compared to clean fuel <0.10.

**Table 1 ijerph-17-05516-t001:** Characteristics according to cooking fuel category (n = 22,118 observations from n = 6671 women, median observations per woman 3).

Demographic and Socio-Economic Characteristics, and Lifestyle Risk Factors	Clean Cooking Fuel (n = 7114)	Coal Cooking Fuel (n = 6811)	Wood Cooking Fuel(n = 8193)
Age (years)	44.84 ± 13.86	44.21 ± 14.87	44.27 ± 14.53
Annual total household income (yuan inflated to 2011)	35,678.59 ± 41,832.66	19,103.19 ± 25,645.52	17,580.39 ± 2122.14
Community urbanicity index	61.97 ± 16.70	47.30 ± 14.14	41.57 ± 11.65
Educational attainment			
No education	1057 (14.86)	2147 (31.52)	2527 (30.84)
Some primary school	1463 (20.57)	1718 (25.22)	2165 (26.42)
Completed primary school	863 (12.13)	735 (10.79)	1197 (14.61)
Some high school	2463 (34.62)	1717 (25.21)	1873 (22.86)
Completed high school or higher	1268 (17.82)	494 (7.25)	431 (5.26)
Average time spent cooking per week (hours)	7.86 ± 6.70	10.05 ± 8.16	10.09 ± 7.78
Ever smoked	293 (4.12)	189 (2.77)	477 (5.82)
Alcohol intake			
Not a consumer	6450 (90.67)	6166 (90.53)	7480 (91.30)
Consumer	664 (9.33)	645 (9.47)	713 (8.70)
Body mass index (kg/m^2^)	23.17 ± 3.54	22.47 ± 3.19	22.13 ± 3.09
Systolic Blood Pressure (mm Hg)	117.27 ± 16.68	115.15 ± 17.61	115.79 ± 17.23
Diastolic Blood Pressure (mm Hg)	76.24 ± 10.29	74.72 ± 10.70	74.92 ± 10.60
Salt (g/1000 kcal/day)	4.98 ± 4.09	5.45 ± 4.27	5.56 ± 4.50
Vegetables (g/1000 kcal/day)	136.38 ± 76.55	142.68 ± 83.92	137.20 ± 79.61
Seafood (g/1000 kcal/day)	16.76 ± 26.38	7.86 ± 19.61	9.00 ± 19.50
Fruit (g/1000 kcal/day)	25.74 ± 60.04	7.83 ± 34.05	9.60 ± 43.87
Meat (g/1000 kcal/day)	36.61 ± 31.12	22.93 ± 27.26	16.81 ± 22.57
Oil (g/1000 kcal/day)	15.31 ± 11.64	9.08 ± 10.52	12.97 ± 10.50
Rice (g/1000 kcal/day)	309.97 ± 180.63	325.23 ± 230.44	366.14 ± 232.73

Values presented are mean ± SD or n (%).

**Table 2 ijerph-17-05516-t002:** Estimated change in predicted blood pressure per year of age for women <40 years of age comparing clean fuel to coal or wood from models with and without dietary intake and proportion mediated by diet (n = 8915 observations from n = 4291 women, median observations per woman 3).

Outcome	Coal			Wood		
	Without Diet	With Diet	% Mediation	Without Diet	With Diet	% Mediation
Change in SBP (mm Hg) by year of age	0.16 (0.05, 0.28)	0.16 (0.05, 0.28)	0.11%	0.04 (−0.09, 0.16)	0.03 (−0.09, 0.16)	0.67%
Change in DBP (mm Hg) by year of age	0.08 (0.00, 0.17)	0.08 (−0.01, 0.17)	0.22%	0.01 (−0.08, 0.11)	0.01 (−0.08, 0.11)	0.85%

All models controlled for the following covariates: fuel category, age (years), fuel category*age, survey year (categorical, reference = 2011), survey year*age, baseline income quintiles (categorical, reference = 3rd quintile), baseline income quintiles*age, urbanicity index quintiles (categorical, reference = 3rd quintile), urbanicity index*age, education level (categorical, reference = “some primary level education”), education level*age, alcohol intake (consumer/not consumer, reference = not consumer), alcohol intake*age, ever smoked (yes/no, reference = no), ever smoked*age, baseline BMI (kg/m^2^), and baseline BMI*age. Models with diet included the following additional covariates: fruits (g/1000 kcal/day), vegetables (g/1000 kcal/day), and seafood (g/1000 kcal/day).

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
