# Peer review of "The Association of Cooking Fuel Use, Dietary Intake, and Blood Pressure among Rural Women in China"

_ijerph, 2020, doi:10.3390/ijerph17155516_

Round 1

Reviewer 1 Report

The authors addressed my and the other reviewers' comments adequately, so I recommend it for publication. In particular, the sensitivity analysis excluding secondary solid fuel use is a helpful and interesting addition that strengthens the paper.

Author Response

Reviewer 1:

The authors addressed my and the other reviewers' comments adequately, so I recommend it for publication. In particular, the sensitivity analysis excluding secondary solid fuel use is a helpful and interesting addition that strengthens the paper.

Response: We appreciate the feedback we received and believe that our paper was strengthened as a result of your inputs. Thank you!

Reviewer 2 Report

In general, the authors have addressed the reviewer comments well and the paper is improved.

I have just a few more follow-up points in relation to the dietary analysis methods: “Unfortunately, we were unable to perform these calculations as these methods have not been developed for China. The NCI method requires collection of FFQ and 24-hour recalls concurrently which was not collected in this study.”

If I am understanding the response correctly, this is not true. The methods referenced for accounting for measurement error in 24-hour recall analysis are statistical models and do not need to be developed separately for each country. The NCI method includes a ‘suite’ of tools, and while analysts can combine FFQ and 24hr if they wish, this is not a requirement (i.e., the NCI macros can be applied to 24h recall data only to estimate diet-health relationships). Not using these methods would be expected to bias the results toward the null. Given the low intakes of the selected foods, I am not sure this would affect the conclusions of the paper.

It is good to see this point addressed in the discussion, but the rationale is not correct. A 3-day recall does not represent an individual’s usual intake because individuals (typically) eat different foods on different days, and so true long-term intake for an individual cannot be estimated with 3 days (season is important but only a part of the reason). Suggest revising this point to better reflect the rationale for use of measurement error models for this analysis, and commenting on how (if) it might impact the results.

“Moreover, while diet was assessed 338 through rigorous methods, a three-day dietary recall may not represent an individual’s long-term 339 usual intake as seasonality is not accounted for in these measures.”

Two other minor points:

  1. Could the authors provide the method for blood pressure measurement i.e auscultation vs. a digital device?
  2. Line 268-269: Could the authors clarify what they mean by "traditional higher BMI"?

Author Response

Reviewer 2:

In general, the authors have addressed the reviewer comments well and the paper is improved.

I have just a few more follow-up points in relation to the dietary analysis methods: “Unfortunately, we were unable to perform these calculations as these methods have not been developed for China. The NCI method requires collection of FFQ and 24-hour recalls concurrently which was not collected in this study.”

If I am understanding the response correctly, this is not true. The methods referenced for accounting for measurement error in 24-hour recall analysis are statistical models and do not need to be developed separately for each country. The NCI method includes a ‘suite’ of tools, and while analysts can combine FFQ and 24hr if they wish, this is not a requirement (i.e., the NCI macros can be applied to 24h recall data only to estimate diet-health relationships). Not using these methods would be expected to bias the results toward the null. Given the low intakes of the selected foods, I am not sure this would affect the conclusions of the paper.

It is good to see this point addressed in the discussion, but the rationale is not correct. A 3-day recall does not represent an individual’s usual intake because individuals (typically) eat different foods on different days, and so true long-term intake for an individual cannot be estimated with 3 days (season is important but only a part of the reason). Suggest revising this point to better reflect the rationale for use of measurement error models for this analysis, and commenting on how (if) it might impact the results.

“Moreover, while diet was assessed 338 through rigorous methods, a three-day dietary recall may not represent an individual’s long-term 339 usual intake as seasonality is not accounted for in these measures.”

Response: We really appreciate the reviewer’s comments regarding the NCI method, which has led us to spend considerable time reviewing the approach and discussing it in light of this particular study. Given that diet was a mediating factor, not the main exposure of interest, and that the effect for which we were exploring mediation (HAP à BP) was relatively small, on top of the point rightly made by the reviewer that intakes of the selected foods are low in this population and thus adjustment is unlikely to have a substantial impact, we do not think that this approach would change the findings or conclusions of the paper. However, we have taken the comment to heart, and have expanded the discussion of this in the revised paper, providing relevant references for readers interested in pursuing this in future research. We do feel that this point is important to make as more researchers start to explore the intersection of dietary intake and environmental exposures. Thank you!

The revised text now reads as follows, starting on line 338:

“Moreover, while diet was assessed through rigorous methods to reduce bias in the measurement of the amount of food consumed on a consumption day, three days of 24-hour recalls may not represent an individual’s usual intake, particularly for foods that are consumed episodically (for example, once a week or only a couple times a month). Model-based approaches such as the National Cancer Institute (NCI) method have been developed to help estimate usual intake of episodically-consumed foods using only a few days of 24-hour recalls [1]. For episodically consumed foods, the NCI method involves a 2-part model in which the probability of consumption is first estimated, and then the amount. The amount part of the model only includes participants with intakes >0. When a limited number of participants have intakes >0, the adjusted usual intake distribution is likely to be imprecise [2]. For foods with a large number of zero intakes, such as those included in this analysis (seafood, fruit and vegetables), including mean estimates from a food frequency questionnaire (FFQ) as covariates substantially improves the power to detect relationships [3]. Unfortunately, however, CHNS did not administer an FFQ. Taking all of these factors into consideration, as well as the fact that diet accounted for a negligible amount of the effect of cooking fuel on SBP in mediation analyses (<1%), implementing the NCI method in this analysis would be unlikely to influence the findings or conclusions. Nonetheless, future studies at the intersection of dietary intake and environmental health should consider the implications of this source of measurement error for their research questions and in their particular sample populations.”

  1. Tooze, J.A.; Midthune, D.; Dodd, K.W.; Freedman, L.S.; Krebs-Smith, S.M.; Subar, A.F.; Guenther, P.M.; Carroll, R.J.; Kipnis, V. A New Statistical Method for Estimating the Usual Intake of Episodically Consumed Foods with Application to Their Distribution. J. Am. Diet. Assoc. 2006, 106, 1575–1587, doi:10.1016/j.jada.2006.07.003.
  2. Batis, C.; Aburto, T.C.; Sánchez-Pimienta, T.G.; Pedraza, L.S.; Rivera, J.A. Adherence to Dietary Recommendations for Food Group Intakes Is Low in the Mexican Population. J. Nutr. 2016, 146, 1897S-1906S, doi:10.3945/jn.115.219626.
  3. Carroll, R.J.; Midthune, D.; Subar, A.F.; Shumakovich, M.; Freedman, L.S.; Thompson, F.E.; Kipnis, V. Taking Advantage of the Strengths of 2 Different Dietary Assessment Instruments to Improve Intake Estimates for Nutritional Epidemiology. Am. J. Epidemiol. 2012, 175, 340–347, doi:10.1093/aje/kwr317.

Two other minor points:

  1. Could the authors provide the method for blood pressure measurement i.e auscultation vs. a digital device?

Response: Thank you for highlighting that this information was missing from the paper. Auscultation was used. We have revised the paper on lines 114-115 to include: “Physicians with additional training measured blood pressure of participants using mercury sphygmomanometers.”

  1. Line 268-269: Could the authors clarify what they mean by "traditional higher BMI"?

Response: Thank you for catching this error. We have revised the manuscript to read “and higher BMI” on line 268.

Reviewer 3 Report

This is an interesting paper and addresses several gaps in HAP studies.

Author Response

Reviewer 3:

This is an interesting paper and addresses several gaps in HAP studies.

Response: Thank you for taking the time to review our manuscript. We appreciate the feedback we received and believe that our paper was strengthened through the peer-review process.

This manuscript is a resubmission of an earlier submission. The following is a list of the peer review reports and author responses from that submission.

Round 1

Reviewer 1 Report

This is an important paper using a retrospective cohort design to investigate the relationship between blood pressure and its association with use of cooking fuels (biomass, coal and clean fuels) among a large sample of Chinese women.

HAP and is a very significant factor in CVD and as the authors point out, the cost of treating CVD by 2030 will be nearly double the cost compared 2012 in Shanghai, emphasising the significance of the issue. 

The paper is methodologically sound and well written and I would recommend it for publication. My only suggestion would be that the authors provide a little more information in the discussion about the prevalence of stove stacking. Is this common among Chinese women using cleaner fuels? And also perhaps they could comment on any estimates of ambient air pollution?  

I am not a statistician. Whilst the approach to modelling seemed appropriate and well executed, someone with greater statistical expertise than I may be able to comment more meaningfully on the methods used. 

Reviewer 2 Report

In this study, Bellows et al. investigated how the relationship between fuel use and blood pressure is impacted by fuel use patterns and risk factors, including diet. These findings contribute to the growing body of epidemiological evidence linking HAP to cardiovascular disease. The paper is well-written and has high-quality figures, but there are a few places where the discussion needs clarification or could be expanded upon.

Abstract: Would be good to include DBP results and differences observed between wood and coal.  

Lines 52-54: Revisit this in the discussion. Thoughts on why effects were greater among younger women than among older women in this study, in contrast to the literature discussed?

Lines 70-71: Suggest listing the cohort provinces here or adding the study map as a supplemental figure

Line 74: Why were men excluded from the analysis? (In contrast to many other fields of research, HAP studies focus overwhelmingly on women, and men’s HAP exposures could provide additional information on how much outdoor air pollution contributes to exposures.)

Lines 85-86: Did the household survey assess any secondary fuel use? Is anything known about the types of stoves participants were using (e.g. traditional versus “improved”).

Line 112: Was current smoking status also assessed?

Lines 121-126: Did your models account for possible correlations among repeated measures?

Lines 166-168: Is “Less than Primary School” in Table 1 equivalent to “Some primary school” as referred to on lines 132, 186, and 211?

Also, parentheses missing from % of alcohol non-consumers among coal and wood users

Line 191: Can you include data from women >= 40 years of age as well in Table 2?

Line 203/Figure 3: Consider making the y-axis scale the same as, or at least closer to, the scale in Figure 2. I understand that the magnified scale makes the differences in slopes clearer, but this is misleading because the absolute difference in DBP is smaller than SBP, and (most importantly) the only statistically significant difference in slopes was for SBP between coal and clean fuels, not DBP. Adding a visual representation of uncertainty might help as well.

Discussion (general): Can you comment on the differences between wood and coal users? Differentiating between those is a strength of this study, as many studies track only use of solid fuel including both wood and coal, so it merits further discussion.

Lines 223-226: This paragraph was confusing initially because the first finding seemed to contradict the previous statement that HAP likely has a direct effect on blood pressure. However, after reading ref. 30, I understand how the results of this study are consistent with other studies and do support that conclusion. To clarify, I would recommend beginning this paragraph with by directly explaining how although unadjusted mean BP was actually lower for solid fuel users than for clean fuel users, after accounting for characteristics and risk factors, solid fuel use was associated with greater increases in BP over time (or something to that effect).

Lines 224-225: Were SBP and DBP differences between solid fuel users and clean fuel users statistically significant?

Line 226: Isn’t salt a CVD risk factor as well? (Though in this case, since salt consumption was slightly higher for solid fuel users, it would temper the effects of other risk factors mentioned here…)

Lines 229-230: Note which of these differences were statistically significant

Line 233: Citation formatting (author-date instead of number)

Supplemental Figure 1: Typo – “urbanicity”

Reviewer 3 Report

The objective of this paper was to assess the relationship between cooking fuel and blood pressure using data from rural women in the CHNS cohort, and assess whether any relationship is mediated by intake of selected foods. The topic is relevant given the global rise in hypertension and other non-communicable disease, and the paper is well-written and clear. There are some questions about how the dietary data were collected and analyzed; these make it difficult to interpret the results of the mediation analysis. Specific comments are below.

Major comments

  1. In general, it would be helpful to include more information about how the dietary intake data were collected and analyzed. For example, the timing of dietary data collection was not clear: is it correct that three 24-h recalls were collected from each person during each data collection round? How were recipe/mixed dish data collected, and how was portion size estimation handled?
  2. Was information on dietary supplements captured and included in the analysis? If not, this should be included as a limitation?
  3. Regarding the selected food groups to assess, there is quite a lot of variability in omega-3 content by type of seafood, and cooking method could impact the net effect on blood pressure. If the specific hypothesis is related to omega-3 consumption, was it possible with available data to further disaggregate seafood intake by cooking method and/or omega-3 content?
  4. Regarding the analysis, how did the authors handle data from each of the 3 dietary recalls in the model? Were each of the days of data included, or do the variables in the model represent the average of the 3 recalls at each time point? Examining diet-health relationships with 24h recall data requires attention to the role of measurement error in estimating usual dietary intake (as described here: https://epi.grants.cancer.gov/events/measurement-error/), but it is not clear how this was handled in the analysis.
  5. Finally, the dietary components examined in the mediation analysis (fruits, vegetables, seafood) may be correlated with each other. Was multicollinearity examined in the models?
  6. In the discussion, the authors do not discuss their results in the context of their first hypothesis. The hypothesis suggests a particular focus on older women, yet the difference among cooking fuel groups in SBP was more apparent among younger women. What is the authors’ interpretation of this finding?
  7. Although the authors looked coal and wood separately, they focused more on coal and based their conclusions on coal alone. It would be helpful to clarify the original rationale for examining 3 groups, and comment in the discussion on the implications for the results related to wood fuel.
  8. It appears a substantial proportion of respondents had missing blood pressure measurements (13,686 out of 37,552 women not taking hypertensive drugs, or ~36%). Given this, it would be helpful to include some information on potential differences in relevant covariates between individuals with and without blood pressure measurements.

Minor comments

  1. As the authors note, measurement of intensity of exposure is a limitation. The same could be added for covariates such as smoking.
  2. Line 24-28: The first 2 sentences have similar information as the last sentence (hard to distinguish the results). Suggest putting all the information together instead of stating separately.
  3. Line 143: Correct the age ranges to show that only one contains 40 years
  4. Table 1: Alcohol intake (not a consumer variable) results are missing the parenthesis in wood and coal column. For descriptive purposes, some estimates seem more precise than necessary (for example, 2 decimal places for mean age and income).
  5. Table 2: Specify what the values in the table represent, and what covariates were included.

Reviewer 4 Report

Abstract

Line 14-15: At the same time, household air pollution from solid fuel cooking fuels continues to affect 600 million Chinese and is also associated with high blood pressure. To me this sound to be more of a finding not a background statement. The study itself looked at cooking fuels, dietary intake and high blood pressure. The author may consider re-writing this sentence.

Line 20 remove the less than sign. When you talk about a range it must be between two values i.e. one low and the other high extreme.

In the abstract I do not see findings related to the contribution of dietary intake on blood pressure change. This need to be specified.

Remove blood pressure on the list of keyword since it is on the study title.

Introduction

Line 36 and 37, the use of coal and wood is prevalent in China for cooking. Similarly, in Africa but what about heating? Does Chinese population use a different fuel source for heating? If so, please add a sentence or otherwise just indicate that coal and wood are used significantly for cooking and space heating.

Line 50-52 requires editing attention; the sentence does not read well. The sentence is also too long maybe the author may consider breaking down sentences according to themes. One sentence, one message.

The aim of the study as described seems to be written like the study objectives. Please consider revising. Please indicate the significant of this study. What knowledge gap does this study seek to address.

Method section

Line 88 on the list of fuel source use, the author ends the sentence by other, which implies that there is a single fuel category which is not listed. If so why the author just not provide the fuel name/ category.

Line 97 what does the acronym MSG stands for?

Line 100, it will be of interest to know what do you mean by after resting. A sentence prior is needed to provide a clear understanding on the issue relating to resting.

Results section

The results are comprehensively reported.

In figure 2 A I see an increase of SBP for coal and wood above 30 years, Same as in Fig 2 B above 60 years. An increase on SBP is seen even for clean fuel. Can the author explain this and indicate what might be the cause on an increase since in the discussion only coal use is indicated as a cause? Results presented in figure 2 need to be cited in text before the figure in order to provide more insight to the reviewer.

 Conclusion

The conclusion section requires additional 2 to 3 sentences indicating or suggesting future prospects in dealing with HAP and blood pressure.

Overall

 The paper was structured with high level of scientific basis and relevance to the field of exposure and health. It was a great feeling to read this paper and also to congratulate authors in tiring up household energy source, dietary and health outcome indicator. Most of the studies in this field are only limited to emission characterisation with little emphasis on the human element.